# Topological fine structure of an energy band

Hui Liu,[1] Cosma Fulga,[2,3] Emil J. Bergholtz,[1] and János K. Asbóth[4,5]

[1]*Department of Physics, Stockholm University, AlbaNova University Center, 106 91 Stockholm, Sweden*
[2]*Leibniz Institute for Solid State and Materials Research,*
*IFW Dresden, Helmholtzstrasse 20, 01069 Dresden, Germany*
[3]*Würzburg-Dresden Cluster of Excellence ct.qmat, 01062 Dresden, Germany*
[4]*Department of Theoretical Physics, Institute of Physics,*
*Budapest University of Technology and Economics, Műegyetem rkp. 3., H-1111 Budapest, Hungary*
[5]*HUN-REN Wigner Research Centre for Physics, H-1525 Budapest, P.O. Box 49., Hungary*

A band with a nonzero Chern number cannot be fully localized by weak disorder. There must remain at least one extended state, which "carries the Chern number." Here we show that a trivial band can behave in a similar way. Instead of fully localizing, arbitrarily weak disorder leads to the emergence of two sets of extended states, positioned at two different energy intervals, which carry opposite Chern numbers. Thus, a single trivial band can show the same behavior as two separate Chern bands. We show that this property is predicted by a topological invariant called a "localizer index." Even though the band as a whole is trivial as far as the Chern number is concerned, the localizer index allows access to a topological fine structure. This index changes as a function of energy within the bandwidth of the trivial band, causing nontrivial extended states to appear as soon as disorder is introduced. Our work points to a previously overlooked manifestation of topology, which impacts the response of systems to impurities beyond the information included in conventional topological invariants.

*Introduction* — Anderson localization [1, 2] is the quantum phase transition across which increasing the disorder leads to the extended states of a disordered system to acquire finite localization lengths, and a conductor becomes insulating. Initially, the scaling theory of localization [3, 4] predicted that such a transition would not occur in two-dimensional (2D) systems that lack any symmetry, i.e., class A of the Altland-Zirnbauer classification [5]. Instead, all states become localized as soon as disorder is introduced, and a thermodynamically large 2D system is insulating even at an infinitesimal disorder strength.

The discovery of the quantum Hall effect [6] and the subsequent development of the theory of topological phases of matter [7–9] changed this paradigm. It was realized that robust extended states are indeed possible in 2D, class A, and their presence is a consequence of their topologically nontrivial character. Thus, for a 2D system in class A, a band with nonzero Chern number cannot be fully localized by weak disorder [10]. Instead, at least one extended state that "carries the Chern number" must remain [11]. Using terminology introduced by Laughlin [10], extended states at different energies and carrying opposite Chern numbers "levitate" towards each other, eventually "annihilating" in order to produce a trivial Anderson insulator. In contrast, trivial bands are expected to fully localize even for infinitesimal disorder strength, provided that the disorder is sufficiently generic [2, 12–22]. These two types of behavior are shown schematically in Fig. 1(a).

Here, we revisit Anderson localization of trivial bands of 2D systems, class A, showing bands which cannot be localized by weak, generic disorder, even though their Chern number vanishes. Instead, there are multiple, robust extended states, which carry opposite Chern numbers, and which, as disorder is further increased, participate in the levitation and annihilation process [see Fig. 1(b)]. This behavior is still due

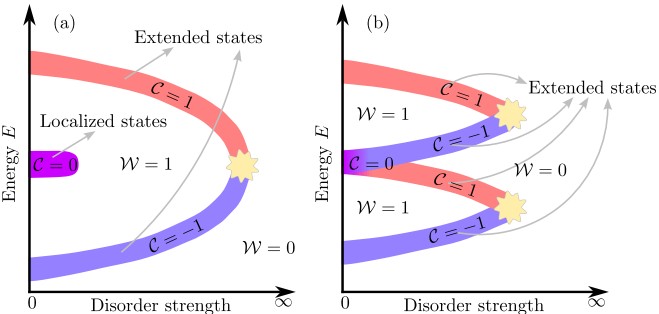

Figure 1. Panel (a): The paradigm of Anderson localization for 2D systems in class A. A trivial band (middle, Chern number $\mathcal{C} = 0$) is fully localized by disorder, whereas nontrivial bands ($\mathcal{C} = \pm 1$) lead to robust extended states. The latter eventually meet and annihilate (star) for larger disorder strength. The index $\mathcal{W}$ denotes the number of chiral edge modes present at energies between/outside those of the extended states. Panel (b): Our main result. The trivial band does not fully localize, but splits into two branches of extended states carrying opposite Chern numbers. As we show, this behavior is a consequence of its topological fine structure.

to nontrivial topology, but it is the consequence of an index that is more general than the Chern number – the *localizer index* [23–26]. While the Chern number is a global property characterizing the entirety of a band, the localizer index can be evaluated for different energies within a band, thus providing access to a topological fine structure. As we show, changes of this topological invariant necessarily lead to the formation of extended states that are robust against disorder.

*An example* — We begin by illustrating our general conclusions using a simple 2D model: a two-band Chern insulator coupled to a single, trivial band. The momentum-space

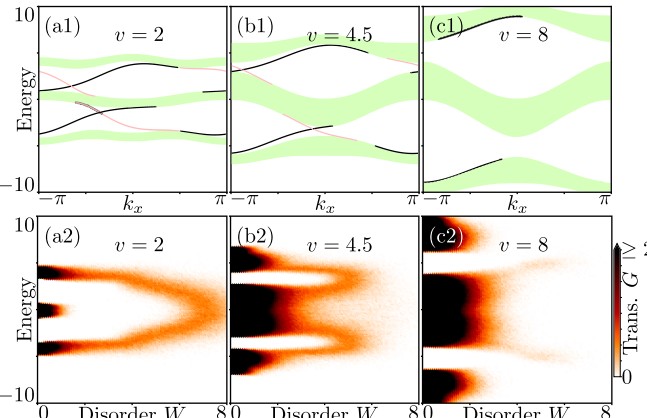

Figure 2. Top panels: band structure of the model in the absence of disorder. We use a ribbon geometry, infinite along the $x$-direction and consisting of 60 unit cells along $y$. Here, we use $p_c = \sum_{i=1}^{N_y/2} |\psi_i|^2$ to label the color, where $0 < p_c < 0.49$ (black), $0.51 \leq p_c \leq 1$ (red), and $0.49 \leq p_c \leq 0.51$ (green) indicate states localized at the top edge, bottom edge, and in the bulk, respectively. Bottom panels: Two-terminal transmission probability, plotted as a function of energy $E$ and disorder strength $W$, for a $60 \times 60$ unit cell system with periodic boundary conditions in the $y$-direction. Darker regions correspond to the presence of states extended on the scale of the finite-size sample, helping us track the levitation and annihilation process. Each point is obtained by averaging over 50 independent disorder realizations.

Hamiltonian reads

$$H(\mathbf{k}) = \begin{pmatrix} h_{11}(\mathbf{k}) & h_{12}(\mathbf{k}) & v \\ h_{12}^*(\mathbf{k}) & -h_{11}(\mathbf{k}) & 0 \\ v & 0 & 0 \end{pmatrix}, \qquad (1)$$

where $\mathbf{k} = (k_x, k_y)$ is the 2D quasimomentum, and the matrix element functions are $h_{11}(\mathbf{k}) = 2(\cos k_x - \cos k_y)$, and $h_{12}(\mathbf{k}) = \sqrt{2}e^{-i\pi/4}(e^{ik_x} + e^{ik_y} + ie^{i(k_x+k_y)} + 1)$. The upper-left $2 \times 2$ block is a Chern insulator with Chern numbers $\mathcal{C} = \pm 1$ for the upper and lower bands [27], whose hopping amplitudes are used to set the unit of energy. The lower right element is the single-band atomic insulator: a trivial flat band with a vanishing onsite potential. It is coupled to the Chern insulator with an amplitude $v$ (see the real space lattice and its corresponding Hamiltonian in the Supplemental Material (SM) [28]).

In the regime of coupling strengths $0 \leq v < 8$, this three-band model has two phases: a topological phase for small coupling, $0 \leq v \lesssim 5.65$, and a trivial phase for large coupling, $v \gtrsim 5.65$. In the topological phase the top and bottom bands have Chern numbers $+1$ and $-1$, respectively, whereas the middle band is trivial, with Chern number 0. In the trivial phase all bands have a vanishing Chern number. These two phases are separated by a topological phase transition, where the gaps between the 1st and 2nd, and the 2nd and 3rd band close simultaneously [see Fig. 2(a1-c1)].

We are interested in the what happens to the energy eigenstates of the model as disorder is turned on and gradually in-

creased. Thus, we add a random onsite potential to the Hamiltonian, different for each of the three orbitals, uniformly distributed in $[-W, W]$, with $W$ denoting the disorder strength. We calculate the two-terminal transmission probability $G$ of a finite-size, square system using the Kwant package [29] in order to numerically estimate the energies at which extended states are present. The leads are attached to the left and right boundaries of the system, and we connect the top and bottom up by periodic boundary conditions along the transverse direction, so as to pick up only the contribution of bulk states to the transmission probability. For additional details of the numerical implementation, see the SM [28] and the code on Zenodo [30].

Localized states' contribution to the transmission probability decays exponentially with system size, whereas for extended bulk states this contribution stays constant, or even grows, as system size is increased. Thus, for a sufficiently large system (in our numerics, $60 \times 60$ unit cells), we can use $G(E, W)$ to locate energies where extended states exist [large transmission probabilities, darker colors in Fig. 2(a2-c2)], and then track the pattern of their levitation and annihilation. Note that in these finite size calculations, localized states with localization length comparable to or larger than the system size also contribute to the conductance, giving significant finite-size conductance values at weak disorder.

When the atomic insulator and Chern insulator are weakly coupled, $v \lesssim 3$, the pattern of levitation and annihilation is unsurprising. As seen in Fig. 2(a2), first the middle, trivial band fully localizes. The outer, nontrivial bands leave behind extended states which levitate towards each other and annihilate around $W \approx 8$. This conventional behavior parallels that observed in earlier works on disordered topological phases [31, 32]. Also for $v \gtrsim 5.65$, the system's behavior is typical: all bands are trivial, they all localize in the presence of disorder, and no levitation and annihilation can be seen.

A qualitatively different behavior, however, can be seen for intermediate coupling, $3.3 \lesssim v \lesssim 5.65$. Here, instead of localizing, the middle band produced two sets of extended states, located symmetrically around $E = 0$. These proceed to levitate away from $E = 0$, and annihilate with the extended states that originated from the top-most and bottom-most bands [see Fig. 2(b2)]. The unconventional extended states emerging from the middle band carry opposite Chern numbers, as we have checked by repeating the calculation with open boundary conditions, and also by computing the scattering-matrix topological invariant [33–35]. Further, by performing three-terminal transport simulations and a finite-size scaling analysis (SM) [28], we have checked that these extended states appear to persist for any disorder strength $W \neq 0$, no matter how small. This, however, does not rule out the possibility that in the thermodynamic limit these extended states only appear at some small but nonzero disorder only, as in the case of the topological Anderson insulator [36, 37]. To rule this out we use appropriate topological invariants below.

*Topological fine structure* — We now show that the phenomenon of extended states emerging from the trivial band,

observed in our example at coupling $v \approx 4.5$, persists also in the thermodynamic limit. For this, we employ a recently introduced tool for computing real-space topological invariants, the "spectral localizer", and quantities computed from it, the localizer index, and the localizer gap [23–26, 38–42]. We rely on revealing what we call the topological fine structure of the middle band, captured by these quantities. We briefly summarize these concepts below, and clarify how they can show robustly delocalized states inside a seemingly topologically trivial (Chern number 0) band. We then calculate these quantities for our model, and prove that for $v \approx 4.5$, there must exist at least two energies in the middle band where eigenstates cannot become localized by weak disorder.

Our starting point is the spectral localizer $\mathcal{L}(r, E)$ [23–26, 38–42], a matrix-valued function, defined for a finite-size sample of the 2D system, class A, with open boundary conditions. It is a continuous function of a reference position $\mathbf{r} = (r_x, r_y)$, which is encoded via a complex number $r = r_x + ir_y$, and of the energy $E$. It is a Hermitian matrix of size $2Nm \times 2Nm$ for a system with $N$ unit cells and $m$ orbitals,

$$\mathcal{L}(r, E) = \begin{pmatrix} H - E & \kappa(X - iY - r^*) \\ \kappa(X + iY - r) & -H + E \end{pmatrix}. \quad (2)$$

Here, $H$ is the Hamiltonian matrix of the finite-size system, $X$ and $Y$ are the matrices of the position operators (more details in the SM [28]). The dimensional parameter $\kappa$ has to be chosen to be small enough, see the SM [28] and Ref. [23] for a more detailed description. In our case and with our units this was fulfilled by setting $\kappa = 0.25$.

The first piece of information provided by the spectral localizer matrix is the so-called localizer index $Q(r, E)$. It is defined as the matrix signature (sig, the number of positive eigenvalues minus the number of negative eigenvalues) of the spectral localizer [24],

$$Q(r, E) = \frac{1}{2}\mathrm{sig}[\mathcal{L}(r, E)]. \quad (3)$$

At energy $E$ in a spectral gap or mobility gap, the localizer index is identical to the number of chiral edge modes at the system boundary for any value of $\mathbf{r}$ chosen deep within the systems' bulk [23–26]. Thus, under these conditions, it contains the same information as the Chern number, giving the value of the Hall conductance [43], and it does not depend on the reference position $\mathbf{r}$, as long as $\mathbf{r}$ is chosen deep in the bulk.

We want to use the localizer index $Q(r, E)$ for energies $E$ inside an energy band; here, its value depends on the reference position $\mathbf{r}$ inside the bulk, but only if there are reference positions where at least one of the eigenvalues of $\mathcal{L}(r, E)$ is 0. It can happen that there are no such values of $\mathbf{r}$, and the localizer index is constant throughout the bulk; under what circumstances this is expected to occur is currently not understood [23–26].

To check whether the localizer index is independent of the reference position, we use the localizer gap $g_\mathcal{L}$ [23–26]. This

is at a given energy $E$ the smallest absolute value of any of $\mathcal{L}(r, E)$'s eigenvalues, i.e. the shortest distance from an eigenvalue of $\mathcal{L}(r, E)$ to zero, at any $\mathbf{r}$ inside the bulk, i.e.,

$$g_\mathcal{L}(E) = \min_{\mathbf{r} \in \text{bulk}} g_\mathcal{L}(r, E); \quad g_\mathcal{L}(r, E) = \min_{\lambda \in \sigma[\mathcal{L}(r,E)]} |\lambda|, \quad (4)$$

where $\sigma[\mathcal{L}]$ is the set of eigenvalues of $\mathcal{L}$. We characterize the topological fine structure of a band using the localizer index $Q(r, E)$, evaluated at energies where the localizer gap $g_\mathcal{L}(E)$ is nonzero, and hence, the index is independent of $r$,

$$Q(E) = Q(r, E) \quad \text{when } g_\mathcal{L}(E) > 0. \quad (5)$$

Besides ensuring the localizer index depends only on energy and not reference position, the localizer gap $g_\mathcal{L}(E)$ also quantifies the robustness of the index. Given Eq. (3), the index $Q(r, E)$ cannot change under any perturbation, unless that perturbation is large enough to close the localizer gap, meaning that $g_\mathcal{L}(E)$ is reduced to zero. This is simply a consequence of Weyl's inequality: when perturbing $H \to \widetilde{H}$ and thus $\mathcal{L} \to \widetilde{\mathcal{L}}$, $Q$ is unchanged as long as $||\widetilde{\mathcal{L}} - \mathcal{L}|| < g_\mathcal{L}$. Here, $||\cdot||$ is the 2-norm of a matrix, the modulus of its largest eigenvalue.

The localizer index $Q(E)$ can reveal topological fine structure of an energy band of a clean 2D system. It predicts the number of chiral edge states that would be seen at $E$ on a large enough sample with open boundary conditions, if weak disorder was added so as to localize the bulk states at energy $E$. This follows from the previous paragraph, with the weak disorder being treated as a perturbation.

If an energy band includes energy values $E_1 < E_2$ where the localizer indices differ, $Q(E_1) \neq Q(E_2)$, then there must exist an energy value between them, $E_1 < E < E_2$, where energy eigenstates remain extended under weak disorder. This is necessarily true, because in the weakly disordered system the only way the number of chiral edge states at $E_1$ and $E_2$ can be different if there is an energy between these values where the bulk gap closes (a topological phase transition in the spectrum of the disordered system). This shows how robustly extended states can emerge from a seemingly topologically trivial band (Chern number 0).

*Topological fine structure in our example* — We now apply the formalism above to our model, the Hamiltonian of Eq. (1), to show that the delocalized states we observe in Fig. 2 are delocalized at weak disorder in the thermodynamic limit. For the numerical work, we took square shaped systems of $20 \times 20$ unit cells with open boundary conditions (we verified that repeating the calculation for larger system sizes does not significantly alter the result). We chose numerous energy values $-10 < E < 10$, and calculated the spectral localizer, Eq. (2), and its eigenvalues of smallest magnitude over a grid of $40 \times 40$ reference positions $\mathbf{r}$ in the central Wigner-Seitz unit cell. Thus we obtained the localizer gap, Eq. (4). We note that, as proven in Ref. [26] and verified by us in the SM [28], for a fixed set of Hamiltonian parameters and $\mathbf{r}$ in the bulk, $g_\mathcal{L}(r, E)$ converges as system size is increased, thus

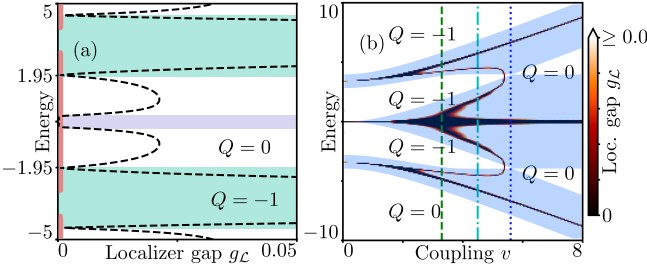

Figure 3.   (a) Topological fine structure of the central energy band at coupling $v = 4.5$: Localizer gap $g_{\mathcal{L}}$ (dashed line, horizontal axis) and localizer index $Q$ (shading and labels). Horizontal gray regions: energy ranges in which the $g_{\mathcal{L}} = 0$, thus $Q(E)$ cannot be evaluated. Vertical red bars on left axis: energy intervals of the bulk bands. (b) Phase diagram of the topological fine structure revealed using the localizer gap $g_{\mathcal{L}}$ (shading in black to yellow/white), obtained from a $10 \times 10$ unit cell system, plotted versus energy $E$ and coupling $v$. Since a change of $Q$ requires a closing of the localizer gap, regions delimited by black lines have uniform $Q$ values, as shown. Semi-transparent light blue shaded areas: energy intervals of the bands. Vertical dashed line: the critical point $v \approx 3.3$ at which the topological phase transition of the middle band occurs with no closing of the bulk gaps – as obtained from the transport simulations. Dash-dotted line: $v = 4.5$, corresponding to panel a). Dotted line: topological phase transition with a change in the Chern numbers, and closing of the bulk band gaps. All plots are with $\kappa = 0.25$ and $g_{\mathcal{L}}$ was computed taking a grid of $40 \times 40$ reference positions $\mathbf{r}$ in the Wigner-Seitz cell.

only the behavior of the system close to the position encoded in $r$ influences $g_{\mathcal{L}}(r, E)$. We picked some $v, E$ values where we also fully diagonalized the spectral localizer to obtain the localizer index $Q(E)$. Since this topological invariant cannot change as long as the localizer gap is nonzero, this numerically costly procedure was only needed at few $(v, E)$ pairs.

First, we show our main result, the robustly delocalized states at $v = 4.5$, in Fig. 3(a). Here, in both spectral gaps we find $Q = -1$, consistent with the Chern numbers of the bands. Remarkably, the localizer gap remains open (e.g. $g_{\mathcal{L}} > 0.02$) even for energies *within* the central band, meaning that the localizer index $Q(E)$ remains well defined for most energies in the band. Moreover, the index changes from $Q = -1$ to $Q = 0$ and then back to $Q = -1$ as energy is scanned through the middle band. Thus there are two topological phase transitions in that band: this is the topological fine structure, predicting two energies where robustly extended states must occur. Here, these transitions are at $E \simeq \pm 1.95$, which agrees with Fig 2(b2).

Second, we show a full phase diagram of the topological fine structure of Hamiltonian, Eq. (1), obtained by repeating the calculation above for many values of $v$, in Fig. 3(b). We find the nontrivial topological fine structure of the middle band throughout the interval $3.3 \lesssim v \lesssim 5.65$. This is consistent with our weak disorder numerics which show robustly extended states in the middle band also when adding additional perturbations to the Hamiltonian, such as shifting the energy of the band (see SM [28]). Note that the mismatch between

the localizer gap closing and the closing of bulk band gaps is a technical issue of choices of $\kappa$, see the SM [28] for more details. Note that the localizer gap is 0 close to $E = 0$ for all values of $v$ we consider, however, there is no topological phase transition in the spectrum here, no robustly extended eigenstates are expected, since the localizer index does not change across $E = 0$.

We emphasize that the emergence of extended states from the trivial band is the consequence of a topological phase transition occurring as $v$ crosses a critical value, between 3.1 and 3.3, as seen in Fig. 3(b) as well as the additional transmission calculation in SM [28]. However, this transition does *not* involve a closing of the bulk gaps of the system, and it is not associated to a change in the Chern numbers of the bands.

*Conclusion and outlook* — We have shown that a seemingly topologically trivial band can exhibit a nontrivial topological fine structure, with a generalization of the Chern number taking different values inside the band. Such a fine structure implies that the band hosts eigenstates that are robustly extended under weak disorder. The generalization of the Chern number capturing the topological fine structure is the localizer index and the localizer gap of the spectral localizer.

Our work will motivate future research in several directions. Saliently, while we have only considered class A in two dimensions, it is an intriguing open question what topological fine structure may exist in other symmetry classes and dimensions. In this context we note that topological markers have recently been generalized to odd-dimensional systems [44].

Another direction worth exploring is the interplay between the fine structure topology and interactions. This, we conjecture, may lead to fractional Chern insulators [45, 46], which usually, but not always [47], require a band with nonzero Chern number. In the present context we envision that stable states may occur at unconventional filling fractions. In particular, Laughlin-like states may instead occur at even denominator band filling (corresponding to odd denominator filling of the effective fine structure band).

On a more practical level, it would be interesting to identify further models exhibiting the topological fine structure phenomenology. Given the generality of our argumentation such models should indeed be ubiquitous. In particular, it would be interesting to find a minimal model, presumably featuring only two energy bands. Nevertheless, we emphasize that the present three band setting is also realistic. In fact, attaching a trivial flat band to a Chern insulator has already been realized in photonic systems [48–50]. We expect that our work will motivate further experimental research in this direction.

Notions of band structure topology have profoundly changed the way we understand phases of matter and altered the paradigm of localization. Topological fine structure provides a natural next level of understanding of these fundamental concepts.

*Acknowledgements* — We thank Ulrike Nitzsche for technical assistance. ICF acknowledges support from the Deutsche Forschungsgemeinschaft (DFG, German Research Foundation) under Germany's Excellence Strategy through the

Würzburg-Dresden Cluster of Excellence on Complexity and Topology in Quantum Matter – *ct.qmat* (EXC 2147, project-ids 390858490 and 392019). HL and EJB were supported by the Swedish Research Council (VR, grant 2018-00313), the Wallenberg Academy Fellows program (2018.0460) of the Knut and Alice Wallenberg Foundation, and the Göran Gustafsson Foundation for Research in Natural Sciences and Medicine. JKA acknowledges support by the National Research Development and Innovation Office (NKFIH) through the OTKA Grant FK 132146.

---

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
