# Peer review of "Topological fine structure of an energy band"

_SciPost Physics_

## Round 1 · Referee Report · Luka Trifunovic (Referee 1) · 2025-12-16

Report
In this manuscript, the authors study 'topological fine structure' which is revealed by introducing a weak disorder, whereupon, in two-dimensional class A systems, all states apart from states at isolated energies E_i localize. The authors study a toy model that exhibits a rather unusual evolution of its 'fine structure', E_i, with increasing disorder strength. Furthermore, the authors propose to compute the values E_i using the method of spectral localizer.
The manuscript contains interesting results and I recommend it for publication after addressing the comments below.
1) A reader would benefit if the statements in the introduction are sharpened. In particular, Anderson localization theory predicts only insulating phase in 2d class A (as opposed to insulating and metallic phases in the same class in 3d). After the discovery of quantum Hall effect, it became clear that not all insulators are the same, hence, it is possible to have insulator-to-(topological) insulator transition. Precisely at this transition, the critical states appear, which are the states labelled "Extended states" on Fig. 1. In the thermodynamic limit, these states sit at isolated energies E_i. At each E_i there are infinitely many critical states: the number of critical states increases with the system size as determined by the critical exponent of the mentioned insulator-to-insulator transition.
I am puzzled by the statement that "trivial bands are expected to fully localize even for infinitesimal disorder" - I do not think such expectation exists in the community. In fact, it is easy to construct a counter-example; Consider a two-dimensional Z2 topological insulator with additional spin rotation symmetry. Upon adding a weak disorder that respects spin rotation symmetry, one finds below the Fermi level both spin-up and spin-down critical states which carry the opposite Chern numbers. If we apply a Zeeman field, the energies of these critical states will split by \Delta_Z. Finally, addition of a most general class-A disorder of strength W (which break spin rotation symmetry) cannot make the critical states disappear as long as W/\Delta_Z is small. In fact, this model has been studied in PhysRevB.85.075115.
In my opinion, the statement that a band with zero Chern number can have a 'topological fine structure' is neither new nor surprising. On other hand, the way the energies E_i of the critical states move with increasing disorder (Fig. 1b), for the model proposed in this manuscript, is rather surprising. Additionally, the idea to use the spectral localizer to numerically determine E_i is novel to the best of my knowledge.
2) Determining the 'topological fine structure' (in class A and 2d) boils down to finding the energies of the critical states E_i for a weak disorder. One can either try to compute E_i by including disorder (expensive) or by doing calculation on the clean system (cheap).
The disordered-system calculation can be done by computing the transmission in a quasi-1d geometry (Kramer-MacKinnon method) or by performing a multifractal analysis. In both cases, one would obtain the results of the same quality (or better) as in Fig. 3b, albeit at a much higher computational and implementation cost.
When it comes to calculation on the clean system there are the two following options:
a) The method proposed in Annals of Physics, 456, 169258, which tells that E_i can be computed by solving the equation:
\sigma_xy^{int}(E_i)=(2*i+1)/2,
where \sigma_xy^{int} is intrinsic contributions to the anomalous Hall conductivity (i.e., an integral of the Berry curvature of the clean system).
The downside of this approach is that the Berry curvature (and thus the intrinsic contributions to the anomalous Hall conductivity) depends on the choice of atomic orbitals of the tight-binding model. In particular, changing the positions of the atomic orbitals within the unit cells may give different solutions for E_i. Hence, after considering all possible positions of the atomic orbitals, this method gives energy windows for each E_i rather than isolated energies.
b) The approach proposed in this manuscript seems to face similar challenges as the above approach. The definition of the spectral localizer explicitly depends on the position operators X and Y. Moving atomic orbitals amounts to changing the diagonal elements of these two operators. For this reason, I expect that the solutions of the equation (g_L is localizer gap)
g_L(E)=0
are intervals in energies rather than isolated energies. In fact, the model studied in the recent preprint confirms this expectation (Fig. 4 in arXiv:2505.09677). Perhaps having such a narrow estimate for E_i is rather exception than a rule?
In any case, a reader would benefit if the authors give a few remarks on how their approach compares to the other approaches.
3) Given the 'topological fine structure' of a model, E_i, it would be interesting to know how these energies evolve and annihilate with increasing disorder W. My expectation is that the critical states closest in energy (and with the opposite topological charges) annihilate first. This expectation does not seem to hold for the model considered in this manuscript. On the other hand, the Fig. 3b looks suspiciously symmetric with respect to E=0 which is not to be expected in class A. Could it be that this unusual behavior of E_i(W) is a consequence of some accidental statistical symmetry present in the model?
4) There is a typo in Fig. 3b, the top most label should read Q=0 instead of Q=-1.
Recommendation
Ask for minor revision

---

## Editorial Decision

unknown